# Participation in the School Milk Program Contributes to Increased Milk Consumption and Dietary Nutrient Intake by Middle School Students in South Korea

**DOI:** 10.3390/nu11102386

**Published:** 2019-10-06

**Authors:** Jee Hyun Lee, Woo Kyoung Kim, Sun Hyo Kim

**Affiliations:** 1Nutrition Education, Graduate School of Education, Dankook University, Yongin City, Gyeonggi-do 16890, Korea; jhlee3265@gmail.com; 2Department of Food Science and Nutrition, College of Natural Science, Dankook University, Cheonan City, Chungcheongnam-do 31116, Korea; wkkim@dankook.ac.kr; 3Department of Technology and Home Economics Education, Kongju National University, Gongju City, Chungcheongnam-do 32588, Korea

**Keywords:** school milk program, nutrient intake, milk, middle school, adolescence, calcium

## Abstract

Milk is considered to be one of the main food sources of calcium for promoting growth and bone health in children and adolescents. This study investigated whether or not participation in a school milk program affected milk consumption and nutrient intake by middle school students in South Korea. In total, 692 middle school students aged 13–16 years old were enrolled in two groups: the school milk program participant group (*n* = 346) and the non-participant group (*n* = 346). The survey examined normal milk consumption status in both groups. The diet record method was applied to analyze the amount of nutrient intake levels. Milk/dairy product consumption was significantly higher in the school milk program participant group for both boys and girls (*p* < 0.001). The school milk program participant group also generally showed higher energy and dietary nutrient intake levels as compared to the non-participant group for both genders (*p* < 0.05). No differences were observed in milk consumption at home or outside school, and calcium intake from animal-derived foods was higher in the school milk program participant group (*p* < 0.001). Therefore, it can be assumed that participating in the school milk program directly resulted in higher calcium intake. Hence, we can report that participating in the school milk program contributes to increased milk consumption and improved the overall nutrient intake.

## 1. Introduction

Consumption of dietary nutrients has a critical impact on the growth and health of adolescents. Since milk contains various essential vitamins and minerals with high bioavailability, consuming milk during adolescence is important for bone growth and general health promotion. Marjo et al. [1] reported that the higher milk and dairy product consumption group was associated with lower consumption of non-alcoholic beverages such as soft drinks, tea, and coffee, and higher fruit and vegetable consumption in children and adolescents. It means that the consumption of milk and dairy product consumption may be a marker for good eating habits. Furthermore, milk is a good food source of calcium in that it provides 276 mg of calcium in a one-cup equivalency [2]. Recent studies have shown that a lack of calcium not only causes well-known calcium deficiency-related diseases such as osteoporosis but also triggers metabolic syndrome, obesity, fractures, hypertension, and dental caries [3,4,5,6,7,8,9]. These results demonstrate that drinking sufficient amounts of milk is important for improvement of general health during adolescence.

In South Korea, the Korean Ministry of Education reported in 2018 that only 30.96% students consumed milk and dairy products daily [10]. According to the 2017 Korean National Health And Nutrition Examination Survey (KHANES), the per day average milk consumption levels in adolescents aged 12 to 18 years in Korea were 191.8 g for boys, 149.0 g for girls, and 171.5 g on average [11]. This indicates that milk consumption in Korean adolescents does not meet the recommend Dietary Guidelines for Koreans, which advocates consuming more than two cups of milk (240 mL/cup) per day for adolescents aged 12 to 18 years [12]. The average dietary calcium intakes per day for adolescents were reported to be 605.5 mg for boys, 436.0 mg for girls, and 524.0 mg on average. Compared to the Dietary Reference Intakes for Koreans (KDRIs), daily calcium intake among Korean adolescents amounted to only 58.9% of the recommended dose, indicating a lack of adequate calcium intake [11,13]. These data indicate that dairy consumption and calcium intake in Korean adolescents are insufficient.

In the 1970s the Korean government initiated the school milk program in elementary schools, which later expanded to middle and high schools and eventually became a nationwide program. In 2018, the implementation rate of the school milk program was 51.1% [14]. The school milk program was started to provide balanced and essential nutrients with milk for growth and health promotion in children and adolescents [15]. A prior study showed that the average daily dietary calcium intake of students who participated in the school milk program was 1.5 times higher than that of students who did not participate [16]. This suggests that participating school milk program maybe one of the ways to increase calcium intake in adolescents. Moreover, teenagers who drank milk during adolescence had a higher preference for dairy and milk products, and continued to consume milk in their adulthood [17].

Research regarding calcium intake in adolescents participating in school milk programs has been widely performed. However, studies on the effects of milk intake from school milk programs on the intake of other nutrients, including calcium, have not been reported. In addition, there is no research including the intake of calcium according to the food group that adolescents consume and the degree of calcium intake of adolescents who participate in the school milk program. Therefore, we undertook this study to examine the effects of the school milk program on nutrient intake by middle school students in South Korea and demonstrate the contribution and positive effects of the school milk program on calcium intake.

## 2. Methods

### 2.1. Study Subjects

This study was designed as a cross-sectional study. In total, 692 subjects (320 boys (46.2%) and 372 girls (53.8%)) aged 13 to 16 years who were studying in middle schools and who voluntarily agreed to take part in the study were enrolled. Of these, 346 subjects (167 boys, 179 girls) who participated in the school milk program were from Cheongju City and Cheonan City, and 346 subjects (153 boys, 193 girls) who did not participate in the school milk program were from Incheon City and Daejeon City in South Korea. All subjects were informed and asked to sign the consent form for the study. The study was conducted from 15 June 2015 to 14 July 2015 and was approved by University Human Research Ethics Committee at Hannam University (approval number 15-01-02-0430).

### 2.2. Questionnaires

The questionnaires consisted of three principal areas: (1) anthropometric characteristics of the subjects, (2) participation status of the school milk program, and (3) daily milk consumption status and reasons for drinking milk. This study was based on self-administered questionnaires, and did not include direct measurements. A total number of 800 questionnaires were collected out of the 800 distributed; the 692 questionnaires which were considered to contain reliable data were included in the study.

### 2.3. Dietary Assessment

The diet record method was applied for examining dietary nutrient intake. Breakfast, lunch, dinner, and snacks over three days (including two weekdays and one weekend day) were assessed. Diet records were self-administered, and subjects were asked to write down what is normally eaten. Dietary data of subjects who recorded eating differently than usual due to events or sickness were excluded from statistical analysis. Subjects were informed and advised in advance regarding the serving size and amount of food by trained dietitians using actual size plastic food models of foods or processed foods. Information regarding lunch meals during weekdays was provided by school dietitians, and the consistency was reviewed and compared with written records of the subjects. Diet records were reviewed and corrected by the dietitians after collection. When diet records were less accurate, they were excluded from statistical analysis so that non-plausible food reports were not included. The computer-aided nutritional analysis program (CAN) pro 4.0 [18] was applied for analysis and assessment of nutrient intake. When the dietary data were being analyzed the subjects with more extreme responses were called to verify that the dietary survey contents were correct so as not to include misreported data.

### 2.4. Statistical Analysis

SPSS (SPSS Inc., Chicago, IL, USA) version 20.0 was used for data analysis. The frequency, percentage, mean, and standard error of mean were calculated from the questionnaires and dietary assessments. The *X*^2^-test or *t*-test was conducted to compare anthropometric characteristics, milk consumption status, percentages of dietary nutrient intake compared to KDRIs [13], and percentages of dietary calcium intake by food group between the participant and non-participant groups in school milk program for both boys and girls. ANCOVA was conducted to compare milk and dairy product intake, dietary nutrient intake, and dietary calcium intake by food group between the participant and non-participant groups in school milk program for both boys and girls after adjusting for age and height. Percentages of dietary nutrient intake with respect to KDRIs between these two groups were compared without adjusting for age and height, as KDRIs were established based on age, sex and anthropometric characteristics. The KRDIs of energy and nutrients included the Estimated Energy Requirement (ERR—energy); Recommended Nutrient Intake (RNI—protein, vitamin A, vitamin C, vitamin B_1_, vitamin B_2_, niacin, vitamin B_6_, folic acid, vitamin B_12_, calcium, phosphorus, magnesium, iron, zinc, and copper); and Adequate Intake (AI—fiber, vitamin D, vitamin E, vitamin K, pantothenic acid, biotin, sodium, chlorine, potassium, manganese, iodine, and selenium). Differences between groups were considered statistically significant at *p* < 0.05.

## 3. Results

Table 1 presents the anthropometric characteristics of the study subjects. Comparison of the participant and non-participant groups revealed significant differences in age and height for both boys and girls (*p* < 0.05) as well as weight and Body Mass Index (BMI) for boys only (*p* < 0.05).

Table 2 shows the daily milk consumption status of the school milk program participant and non-participant groups. A significant difference in daily milk/dairy product intake was observed between the groups for both boys and girls (*p* < 0.001), and this difference was more than double in the participant group: non-participant subjects 114.0 ± 12.5 g/d for boys and 127.2 ± 9.8 g/d for girls; participant subjects 337.8 ± 12.0 g/d for boys and 283.1 ± 10.2 g/d for girls. The daily recommended intake of milk for Korean adolescents is more than two cups of milk, and this criterion was satisfied by 39.9% of boys and 13.9% of girls in the non-participant group and 68.9% of boys and 43.0% of girls in the participant group. Since no significant differences were observed for milk consumption at home and outside school, these results indicates that the difference in milk consumption occurred at school as a result of the school milk program.

Table 3 shows the daily nutrients intake levels of the participants and non-participants in the school milk program. Participant groups generally showed significantly higher nutrient and energy intake levels than the non-participant group for both boys and girls. For boys, all nutrients except vitamin E were higher in the participant group (*p* < 0.05). For girls, nutrients except niacin, magnesium, and selenium were higher in the participant group (*p* < 0.05). Additionally, calcium showed a significantly higher intake in both genders between the groups (*p* < 0.001), with values being almost twice as high in the participant group as compared to the non-participant group: 392.5 ± 16.6 mg/d for boys and 408.8 ± 13.5 mg/d for girls in the non-participant group, and 664.5 ± 15.8 mg/d for boys and 622.6 ± 14.1 mg/d for girls in the participant group.

Table 4 shows the percentages of daily nutrient intake compared to KDRIs of participants and non-participants in the school milk program. Boys in the participant group had higher intake percentages than the non-participant group, except for vitamin E and vitamin B1 (*p* < 0.05). Girls in the participant group had higher intake percentages than the non-participant group, except for niacin, biotin, magnesium, copper, and selenium (*p* < 0.05). Boys and girls in the participant group met 71.1% and 73.0%, respectively, of the required calcium intake, whereas boys and girls in the non-participant group met only 36.9% and 43.8%, respectively, of the desired amount. These results indicate that the school milk program participant group generally had a higher nutrient intake percentage, which satisfies the recommended intake.

Table 5 shows the amounts and percentages of daily calcium intake by food group. The participant group had significantly higher contribution of calcium from milk and dairy products for both genders (*p* < 0.001). The non-participant group had 132.0 ± 13.4 mg/d and 144.5 ± 11.0 mg/d for boys and girls, respectively, whereas the participant group had 344.0 ± 12.8 mg/d and 314.9 ± 11.4 mg/d for boys and girls, respectively, showing over 50% contribution from the ‘milk and dairy products’ group Additionally, the school milk participant groups had significantly higher animal-derived calcium intakes, showing more than 60% contribution in both genders (*p* < 0.001). This indicates that the high calcium intake in participation groups was due to milk consumption from the school milk program.

## 4. Discussion

This study examined dietary nutrient intake by considering participation of middle school students in South Korea in a school milk program. We additionally identified the contribution and impact of the school milk program on milk consumption and calcium intake.

Our study data revealed that groups participating in the school milk program showed significantly higher milk/dairy product intakes (*p* < 0.001), whereas there were no significant differences in milk consumption at home or outside school. Overall, after adjusting for age and height, the participant groups showed a higher nutrient intake than non-participant groups for both genders. The school milk participant group in particular showed a higher calcium intake than the non-participant group (*p* < 0.001). In comparing the percentage of daily calcium intake by food group, the participant group showed a significantly higher animal-derived calcium intake for both boys and girls (*p* < 0.001), and the ‘milk/dairy product’ group was the prominent food source of calcium, exceeding 50% of calcium intake. Studies have been conducted in South Korea to examine the benefits of participating in school milk programs in addition to school lunch programs. Kim et al. [16] assessed students from 52 schools implementing the school milk program as well as 32 schools that did not implement the same program. They reported that students who had lunch with milk provided by the school milk program showed an overall higher nutrient intake. Lee et al. [20] demonstrated that the main route for students to get milk was through the school milk program, and the percentage of milk consumption was significantly higher in students enrolled in this program. These previous studies indicate a similarity with the present study, where study subjects participating in the school milk program showed higher intakes of milk and dairy products. Korean adolescents consumed less than their daily recommended milk intake. In addition, the higher milk intake of adolescents who participated in school milk program compared to adolescents who did not participate shows that school milk program plays a favorable role in adolescent milk intake.

Other studies have investigated the association of nutrient intake and nutritional status, with milk and dairy consumption in adolescents. Keast et al. [21] investigated the 2005–2008 National Health and Nutrition Examination Survey (NHANES) of children aged 8 to 18 years, and reported that high dairy consumption is associated with high calcium intake. In evaluating data from the 2003–2006 NHANES, Nickals et al. [22] reported that white milk contributed to 29% of calcium intake, after studying 7332 subjects aged 2 to 18 years. As a result of the present study, in boys and girls who did not participate in the school milk program, the calcium intakes from milk/dairy products were 132.0 ± 13.4 mg/d and 144.5 ± 11.0 mg/d, respectively, which accounted for 24.39% and 29.69% of total calcium intakes, respectively. However, same subjects who participated in the school milk program consumed more than 50% of their daily calcium intake from school-provided milk (344.0 ± 12.8 mg/d for boys and 314.9 ± 11.4 mg/d for girls), which indicates that milk intake in school is a very important source of calcium in adolescence. In a study in Sydney, Australia enrolling 222 children aged 8 to 10 years, Rangnan et al. [23] showed that the high dairy intake group not only showed higher intakes of energy, protein, and nutrients but also diets of higher nutritional quality. Kobayashi et al. [24] from Japan also reported that both primary and junior high school students who ate school lunches with milk showed significantly higher calcaneal bone mass than students who did not consume milk; the former also showed higher milk and dairy product intakes. Our current study also showed that the school milk program participant groups consumed more milk and dairy products and had higher nutrient and calcium intakes, which corroborates the results from the previous studies.

After studying 127 subjects aged 9 to 18 years from the 2001–2002 NHANES, Gao et al. [25] reported that it is hard to obtain sufficient dietary calcium intake without eating dairy food. Fayet et al. [26] suggested that the group consuming milk was more likely to satisfy the dietary calcium Estimated Average requirements (EARs), after studying Australian adolescents. Murphy et al. [27] also indicated that the milk-drinking group showed higher milk consumption than the non-milk-drinking group, along with a higher dietary nutrient intake, in adolescents in the United States. Furthermore, Coudray [28] analyzed the 2005–2007 French National Food Consumption Survey 2 and reported that dairy products were strong contributors to calcium intake, with dairy products and milk being the main sources of calcium in subjects aged 3 to 17 years. After analyzing 5876 subjects from the NHANES 2011–2014, O’Neil et al. [29] reported that milk was the top ranked food source for calcium among all age groups. This is similar to the result from the present study, where milk was found to be the main contributor to calcium intake, constituting over 50% of the calcium requirement from milk and dairy food groups. Compared with calcium of KRDIs for Korean boys and girls, calcium intake of the present study was low at 36.9% (boys) and 43.8% (girls) in subjects who did not participate in school milk program, but was significantly high at 71.1% (boys) and 73.0% (girls) for school milk program participants, which suggested that school milk program improved calcium intake of subjects (*p* < 0.001). This study confirms the importance of drinking milk to increase calcium intake in adolescents.

Some studies have further reported that not only plain milk but also flavored milk contributes toward increasing milk consumption and nutrient intake. In a study of 53 cases, Fayet-Moore [30] reported that children favored flavored milk, which received a higher palatability score than plain milk. It further stated that children drinking flavored milk showed higher milk consumption and were inclined to drink less plain milk during unavailability of flavored milk, consequently leading to lower consumption of milk. Another study by Fayet et al. [26] reported that the subjects drinking milk (including flavored milk) showed a significantly higher total milk consumption, and that the flavored milk drinkers were more likely to meet EARs for calcium (1.7 times more than plain milk drinkers). This indicates flavored milk to be an alternative option to promote milk consumption, with no significant effect on body mass index, waist circumference, or physical activity. It was also reported by Henry et al. [31] that total milk intake decreased by 12.3% when chocolate milk was removed from school milk programs for children in Saskatoon, Canada. This strongly supports the impact of milk flavor on milk consumption by children. The school milk program in Korea currently provides only whole milk and no flavored milk. Therefore, if flavored milk is added to the school milk program in South Korea, we expect milk and calcium intakes to increase. However, since information regarding the association between flavored milk consumption and energy and sugar intake is lacking, further research needs to be conducted on flavored milk to assess the effects on milk consumption and nutrient intake in the school milk program in South Korea.

The limitations of this study are as follows: (1) The nutrient intake was only investigated by the food record method for three days and the research method for determining the daily food intake was not applied; and (2) The number of subjects was too small for generalization. However, the results of this study suggest that the school milk program can improve the overall nutrient intake of adolescents, providing a basis for the expansion of school milk program. This result can apply to nutrition education, as milk intake helps to balance the nutrient intake necessary for adolescent growth.

In conclusion, participating in the school milk program contributes to an increase in milk consumption, with subsequent improvement in overall nutrient intake in adolescents. Moreover, milk consumption especially contributes to calcium intake. Further studies are required in order to promote the participation and implementation rate of a school milk program among schools in South Korea.

## Figures and Tables

**Table 1 nutrients-11-02386-t001:** Anthropometric characteristics of participants and non-participants in the school milk program in South Korea.

Variables	Boys			Girls		
	Participants (*n* = 167)	Non-Participants (*n* = 153)	Total (*n* = 320)	Participants (*n* = 179)	Non-Participants (*n* = 193)	Total (*n* = 372)
Age (years)	14.1 ± 0.1 ^1,^***^,^^2^	13.6 ± 0.0	13.8 ± 0.0	13.9 ± 0.1 ***	13.5 ± 0.0	13.7 ± 0.0
Height (cm)	168.9 ± 0.6 *	167.0 ± 0.6	168.0 ± 0.4	158.6 ± 0.4 **	160.0 ± 0.4	159.3 ± 0.3
Weight (kg)	59.3 ± 0.9 **	55.9 ± 0.8	57.6 ± 0.6	49.6 ± 0.6 ^NS^	49.5 ± 0.5	49.5 ± 0.4
BMI (kg/m^2^) ^3^	20.7 ± 0.3 *	19.9 ± 0.2	20.3 ± 0.2	19.7 ± 0.2 ^NS^	19.3 ± 0.2	19.5 ± 0.1
Total (*n*(%))	167 (52.2)	153 (47.8)	320 (46.2)	179 (48.1)	193 (51.9)	372 (53.8)

^1^ Mean ± Standard error of mean. ^2^ * *p* < 0.05. ** *p* < 0.01. *** *p* < 0.001. NS: not significant difference (α = 0.05). ^3^ BMI = weight(kg)/height(m^2^). Underweight: <18.5 kg/m^2^, normal weight: 18.5–22.9 kg/m^2^, overweight: 23.0–24.9 kg/m^2^, obese: ≥25.0 kg/m^2^ [19].

**Table 2 nutrients-11-02386-t002:** Milk consumption status of participants and non-participants in the school milk program in South Korea.

Variables	Boys				Girls			
	Participants	Non-Participants	Total	*X*^2^-Test	Participants	Non-Participants	Total	*X*^2^-Test
Daily milk/dairy product intake (g)	337.8 ± 12.0 ^1^	114.0 ± 12.5	230.8 ± 11.3	***^,2^	283.1 ± 10.2	127.2 ± 9.8	202.2 ± 8.5	***
Drink milk at home (*n*(%))								
No	29 (17.4) ^3^	30 (19.6)	59 (18.4)	NS	60 (33.5)	55(28.5)	115(30.9)	NS
Yes	138 (82.6)	123 (80.4)	261 (81.6)		119 (66.5)	138(71.5)	257(69.1)	
Drink milk outside school (n(%))								
No	116 (69.5)	100 (65.4)	216 (67.5)	NS	135 (75.4)	148(76.7)	283(76.1)	NS
Yes	51 (30.5)	53 (34.6)	104 (32.5)		44 (24.6)	45(23.3)	89(23.9)	
Daily milk intake (*n*(%))								
None	0 (0.0)	25 (16.3)	25 (7.8)	***				***
					0 (0.0)	47 (24.4)	47 (12.6)	
<Half a cup	0 (0.0)	13 (8.5)	13 (4.1)		4 (2.2)	35 (18.1)	39 (10.5)	
					7 (3.9)	31 (16.1)	38 (10.2)	
Half a cup	1 (0.6)	18 (11.8)	19 (5.9)		91 (50.9)	53 (27.5)	144 (38.7)	
					58 (32.4)	20 (10.4)	78 (21.0)	
1 cup	51 (30.5)	36 (23.5)	87 (27.2)		19 (10.6)	7 (3.5)	26 (7.0)	
2 cups	78 (46.7)	40 (26.2)	118 (36.9)					
≥3 cups	37 (22.2)	21 (13.7)	58 (18.1)					
Reasons for drinking milk (*n*(%))								
Taste	32 (19.2)	37 (28.9)	69 (23.4)	*	29 (16.2)	55 (37.7)	84 (25.9)	***
Hunger	8 (4.8)	6 (4.7)	14 (4.8)		22 (12.3)	8 (5.5)	30 (9.2)	
For health	24 (14.3)	5 (3.9)	29 (9.8)		20 (11.2)	11 (7.5)	31 (9.5)	
To quench thirst	34 (20.4)	33 (25.8)	67 (22.7)		26 (14.5)	24 (16.4)	50 (15.4)	
To increase height	55 (32.9)	41 (32.0)	96 (32.6)		43 (24.0)	21 (14.4)	64 (19.7)	
For bone health	5 (3.0)	1 (0.8)	6 (2.0)		7 (3.9)	4 (2.7)	11 (3.4)	
Advised to drink	9 (5.4)	5 (3.9)	14 (4.7)		32 (17.9)	23 (15.8)	55 (16.9)	

^1^ Mean ± Standard error of mean after adjusting for age and height. ^2^ * *p* < 0.05. *** *p* < 0.001. NS: not significant difference (α = 0.05). ^3^ Percentage of total.

**Table 3 nutrients-11-02386-t003:** Daily nutrient intake of participants and non-participants in the school milk program in South Korea.

Nutrients	Boys			Girls		
	Participants	Non-Participants	Total	Participants	Non-Participants	Total
Energy (kcal)	2280.7 ± 55.0 ^1,^***^,^^2^	1888.0 ± 57.7	2093.0 ± 39.9 ^3^	1965.1 ± 36.1 ***	1720.1 ± 34.7	1838.0 ± 24.7
Protein (g)	91.4 ± 2.9 ***	74.3 ± 3.0	83.2 ± 2.0	75.3 ± 1.6 **	67.5 ± 1.6	71.2 ± 1.1
Fiber (g)	17.9 ± 0.4 ***	15.4 ± 0.4	16.7 ± 0.3	16.3 ± 0.4 **	14.6 ± 0.3	15.4 ± 0.2
Vitamin A (μg RAE)	858.3 ± 26.7 ***	628.6 ± 28.0	748.5 ± 19.5	782.3 ± 23.2 ***	653.3 ± 22.2	715.3 ± 15.7
Vitamin D (μg/d)	7.4 ± 0.3 ***	3.1 ± 0.3	5.3 ± 0.2	5.2 ± 0.2 ***	3.3 ± 0.2	4.2 ± 0.1
Vitamin E (mg α-TE)	20.4 ± 1.5 ^NS^	17.1 ± 1.6	18.8 ± 1.0	16.1 ± 0.5 **	13.8 ± 0.5	14.9 ± 0.3
Vitamin K (μg)	158.8 ± 7.2 ***	116.5 ± 7.5	138.6 ± 5.1	136.7 ± 5.2 ***	106.6 ± 5.0	121.1 ± 3.6
Vitamin C (mg)	73.2 ± 2.4 ***	48.8 ± 2.6	61.5 ± 1.8	70.6 ± 2.1 ***	56.1 ± 2.0	63.1 ± 1.4
Vitamin B1 (mg)	1.8 ± 0.1 *	1.6 ± 0.1	1.7 ± 0.0	1.5 ± 0.0 **	1.4 ± 0.0	1.5 ± 0.0
Vitamin B2 (mg)	1.8 ± 0.1 ***	1.3 ± 0.1	1.5 ± 0.0	1.5 ± 0.0 ***	1.1 ± 0.0	1.3 ± 0.0
Niacin (mg NE)	18.5 ± 0.6 **	15.9 ± 0.6	17.3 ± 0.4	15.5 ± 0.4 ^NS^	14.8 ± 0.4	15.1 ± 0.3
Vitamin B6 (mg)	1.8 ± 0.1 ***	1.4 ± 0.1	1.6 ± 0.0	1.5 ± 0.0 *	1.4 ± 0.0	1.4 ± 0.0
Folic acid (μg DFE)	451.5 ± 10.2 ***	350.4 ± 10.7	403.2 ± 7.6	409.7 ± 8.9 ***	330.5 ± 8.5	368.6 ± 6.2
Vitamin B12 (μg)	9.0 ± 0.3 ***	5.5 ± 0.3	7.3 ± 0.2	7.3 ± 0.2 ***	5.8 ± 0.2	6.5 ± 0.2
Pantothenic acid (mg)	6.5 ± 0.2 ***	4.8 ± 0.2	5.7 ± 0.1	5.5 ± 0.1 ***	4.5 ± 0.1	5.0 ± 0.1
Biotin (μg)	20.9 ± 0.9 **	17.0 ± 0.9	19.1 ± 0.6	17.9 ± 0.6 **	15.4 ± 0.5	16.6 ± 0.4
Calcium (mg)	664.5 ± 15.8 ***	392.5 ± 16.6	534.4 ± 14.4	622.6 ± 14.1 ***	408.8 ± 13.5	511.7 ± 11.4
Phosphorus (mg)	1298.7 ± 32.1 ***	1009.3 ± 33.6	1160.3 ± 24.0	1115.6 ± 21.3 ***	940.7 ± 20.4	1024.9 ± 14.9
Sodium (mg)	4433.4 ± 114.8 ***	3760.7 ± 120.3	4111.8 ± 81.5	3916.2 ± 91.6 ***	3335.5 ± 87.9	3614.9 ± 62.1
Chlorine (mg)	880.7 ± 58.7 ***	577.8 ± 61.6	735.9 ± 41.7	736.7 ± 60.7 *	562.4 ± 58.2	646.3 ± 40.5
Potassium (mg)	2752.9 ± 52.5 ***	2055.8 ± 55.0	2419.6 ± 41.7	2450.6 ± 46.9 ***	2039.7 ± 45.0	2237.4 ± 33.1
Magnesium (mg)	71.4 ± 2.4 *	64.2 ± 2.5	67.9 ± 1.7	70.3 ± 2.2 ^NS^	70.5 ± 2.2	70.4 ± 1.5
Iron (mg)	14.9 ± 0.3 ***	11.6 ± 0.4	13.3 ± 0.3	12.7 ± 0.3 ***	11.3 ± 0.3	12.0 ± 0.2
Zinc (mg)	12.9 ± 0.3 ***	10.1 ± 0.4	11.6 ± 0.2	10.9 ± 0.2 ***	9.5 ± 0.2	10.1 ± 0.1
Copper (mg)	1058.5 ± 24.8 ***	865.6 ± 26.0	966.3 ± 17.9	945.1 ± 19.0*	891.2 ± 18.2	917.2 ± 12.6
Manganese (mg)	3.3 ± 0.1 ***	2.9 ± 0.1	3.1 ± 0.1	3.1 ± 0.1 **	2.8 ± 0.1	2.9 ± 0.0
Iodine (μg)	469.0 ± 36.0 ***	134.6 ± 37.7	309.1 ± 26.3	746.0 ± 45.3 ***	86.6 ± 43.5	403.9 ± 33.9
Selenium (μg)	123.7 ± 3.4 **	108.1 ± 3.6	116.2 ± 2.4	102.0 ± 2.2 ^NS^	98.6 ± 2.1	100.2 ± 1.4

^1^ Mean ± Standard error of mean after adjusting for age and height. ^2^ * *p* < 0.05. ** *p* < 0.01. *** *p* < 0.001. NS: not significant difference (α = 0.05). ^3^ Mean ± Standard error of mean.

**Table 4 nutrients-11-02386-t004:** Percentages of daily nutrient intake compared to KDRIs of participants and non-participants in the school milk program in South Korea.

Nutrients ^1^	Boys			Girls		
	Participants	Non-Participants	Total	Participant	Non-Participant	Total
Energy	89.9 ± 2.4 ^2,^***^,3^	74.7 ± 1.6	82.6 ± 1.5	98.3 ± 1.8 ***	85.9 ± 1.6	91.9 ± 1.2
Protein	159.8 ± 5.3 ***	133.4 ± 3.7	147.2 ± 3.3	150.4 ± 3.2 **	135.1 ± 3.0	142.5 ± 2.2
Fiber	71.6 ± 1.6 ***	61.6 ± 1.5	66.8 ± 1.1	81.1 ± 1.8 **	73.7 ± 1.5	77.2 ± 1.2
Vitamin A	110.4 ± 3.7 ***	84.4 ± 3.1	98.0 ± 2.6	122.1 ± 3.4 ***	100.7 ± 3.3	111.0 ± 2.4
Vitamin D	75.7 ± 3.3 ***	28.7 ± 1.5	53.2 ± 2.3	54.5 ± 2.2 ***	30.8 ± 1.6	42.2 ± 1.5
Vitamin E	199.8 ± 16.5 ^NS^	169.0 ± 8.7	185.1 ± 9.6	158.3 ± 4.8 **	138.5 ± 4.3	148.1 ± 3.3
Vitamin K	218.3 ± 9.2 ***	166.9 ± 10.4	193.7 ± 7.1	212.4 ± 8.3 ***	162.1 ± 6.8	186.3 ± 5.5
Vitamin C	77.2 ± 2.9 ***	55.4 ± 2.3	66.8 ± 2.0	70.7 ± 2.2 ***	56.7 ± 1.8	63.4 ± 1.4
Vitamin B1	154.0 ± 4.7 ^NS^	143.9 ± 4.1	149.2 ± 3.1	137.1 ± 3.6 *	125.4 ± 3.5	131.0 ± 2.5
Vitamin B2	115.1 ± 3.5 ***	81.6 ± 2.3	99.1 ± 2.3	126.9 ± 3.1 ***	93.5 ± 2.5	109.6 ± 2.2
Niacin	119.6 ± 3.7 **	105.4 ± 3.3	112.8 ± 2.5	103.9 ± 3.1 ^NS^	99.4 ± 2.4	101.6 ± 1.7
Vitamin B6	117.9 ± 4.0 ***	95.0 ± 2.5	106.9 ± 2.5	106.3 ± 2.2 *	99.9 ± 2.4	103.0 ± 1.6
Folic acid	121.0 ± 2.7 ***	98.3 ± 2.8	110.1 ± 2.0	109.9 ± 2.6 ***	93.7 ± 2.1	101.4 ± 1.7
Vitamin B12	374.6 ± 14.1 ***	233.8 ± 11.2	307.2 ± 10.0	319.9 ± 10.6 ***	248.2 ± 9.2	282.7 ± 7.2
Pantothenic acid	131.1 ± 3.7 ***	94.4 ± 2.4	113.6 ± 2.5	109.3 ± 1.9 ***	90.6 ± 1.9	99.6 ± 1.4
Biotin	79.1 ± 3.7 *	68.7 ± 2.6	74.1 ± 2.3	68.3 ± 2.3 ^NS^	62.8 ± 2.0	65.4 ± 1.5
Calcium	71.1 ± 1.9 ***	36.9 ± 1.4	54.7 ± 1.5	73.0 ± 1.9 ***	43.8 ± 1.3	57.8 ± 1.3
Phosphorus	101.1 ± 2.8 ***	76.3 ± 2.0	89.3 ± 1.8	93.6 ± 1.9 ***	77.9 ± 1.6	85.4 ± 1.2
Sodium	295.6 ± 7.8 ***	250.7 ± 7.1	274.1 ± 5.4	259.2 ± 6.1 ***	224.1 ± 5.3	241.0 ± 4.1
Chlorine	39.2 ± 2.6 ***	24.2 ± 2.3	32.0 ± 1.8	32.8 ± 2.5 *	23.7 ± 2.4	28.1 ± 1.8
Potassium	79.1 ± 1.5 ***	58.2 ± 1.5	69.1 ± 1.2	70.5 ± 1.3 ***	57.9 ± 1.2	63.9 ± 0.9
Magnesium	21.2 ± 0.7 ^NS^	19.8 ± 0.8	20.5 ± 0.5	23.6 ± 0.7 ^NS^	24.3 ± 0.7	24.0 ± 0.5
Iron	106.9 ± 2.4 ***	82.4 ± 2.3	95.2 ± 1.8	82.1 ± 1.6 ***	70.2 ± 1.5	76.0 ± 1.2
Zinc	152.3 ± 4.3 ***	125.8 ± 3.2	139.6 ± 2.8	132.7 ± 2.6 ***	118.8 ± 2.4	125.5 ± 1.8
Copper	137.9 ± 3.4 ***	117.4 ± 3.0	128.1 ± 2.4	124.2 ± 2.5 ^NS^	121.1 ± 2.3	122.6 ± 1.7
Manganese	82.5 ± 1.7 **	74.2 ± 1.8	78.5 ± 1.3	87.3 ± 1.9 **	80.2 ± 1.7	83.6 ± 1.3
Iodine	337.8 ± 35.4 ***	128.6 ± 12.5	237.8 ± 20.2	528.1 ± 48.3 ***	109.0 ± 9.1	310.7 ± 26.0
Selenium	202.2 ± 5.8 **	179.3 ± 4.7	191.3 ± 3.8	166.1 ± 3.4 ^NS^	165.7 ± 3.4	165.9 ± 2.4

^1^ Nutrient intake was compared with the reference value of the 2015 Dietary Reference Intakes for Koreans (KRDIs) [13]. KRDIs of energy and nutrients include the Estimated Energy Requirement (ERR—energy); Recommended Nutrient Intake (RNI—protein, vitamin A, vitamin C, vitamin B1, vitamin B2, niacin, vitamin B6, folic acid, vitamin B12, calcium, phosphorus, magnesium, iron, zinc, copper); and Adequate Intake (AI—fiber, vitamin D, vitamin E, vitamin K, pantothenic acid, biotin, sodium, chlorine, potassium, manganese, iodine, selenium). ^2^ Mean ± Standard error of mean. ^3^ * *p* < 0.05. ** *p* < 0.01. *** *p* < 0.001. NS: not significant difference (α = 0.05).

**Table 5 nutrients-11-02386-t005:** Amounts and percentages of daily calcium intake by food group of participants and non-participants in the school milk program in South Korea (mg/d (%)).

Food Group	Boys			Girls		
	Participants (*n* = 167)	Non-Participant (*n* = 153)	Total (*n* = 320)	Participants (*n* = 179)	Non-Participants (*n* = 193)	Total (*n* = 372)
Grains	53.3 ± 4.0 ^1,NS,2^ (8.0 ± 0.5 ^3,^***)	50.8 ± 4.2 (14.2 ± 0.8)	52.1 ± 2.8 ^3^ (11.0 ± 0.5)	57.9 ± 4.6 ^NS^ (9.6 ± 0.8 ***)	53.2 ± 4.4 (13.5 ± 0.8)	56.1 ± 3.1 (11.6 ± 0.6)
Potato starch	3.5 ± 0.3 ** (0.6 ± 0.0 *)	2.5 ± 0.3 (0.8 ± 0.1)	3.0 ± 0.2 (0.7 ± 0.1)	2.2 ± 0.3 ** (0.4 ± 0.0 ***)	3.3 ± 0.3 (0.8 ± 0.1)	2.6 ± 0.2 (0.6 ± 0.1)
Sugar	0.9 ± 0.3 ^NS^ (0.9 ± 0.1 ^NS^)	0.7 ± 0.3 (0.1 ± 0.0)	0.8 ± 0.2 (0.1 ± 0.0)	0.9 ± 0.2 ^NS^ (0.2 ± 0.0 ^NS^)	0.7 ± 0.2 (0.2 ± 0.1)	0.7 ± 0.1 (0.2 ± 0.0)
Beans	23.8 ± 2.1 ^NS^ (3.4 ± 0.3 ***)	24.9 ± 2.2 (7.0 ± 0.4)	24.3 ± 1.4 (5.1 ± 0.3)	27.6 ± 2.0 ^NS^ (4.4 ± 0.3 ***)	27.4 ± 1.9 (7.0 ± 0.4)	27.7 ± 1.3 (5.7 ± 0.3)
Seeds/nuts	4.0 ± 0.3 ** (0.6 ± 0.1 ^NS^)	2.6 ± 0.3 (0.8 ± 0.1)	3.3 ± 0.2 (0.7 ± 0.1)	3.1 ± 0.3 * (0.5 ± 0.0 ^NS^)	2.3 ± 0.3 (0.6 ± 0.1)	2.7 ± 0.2 (0.5 ± 0.0)
Vegetables	99.6 ± 3.4 *** (15.4 ± 0.6 ***)	71.4 ± 3.5 (22.0 ± 0.9)	86.1 ± 2.4 (18.6 ± 0.5)	87.3 ± 2.9 *** (14.4 ± 0.6 ***)	66.0 ± 2.7 (18.5 ± 0.6)	78.8 ± 2.0 (16.5 ± 0.4)
Mushrooms	0.6 ± 0.1 *** (0.1 ± 0.0 ^NS^)	0.2 ± 0.1 (0.1 ± 0.0)	0.4 ± 0.1 (0.1 ± 0.0)	0.1 ± 0.0 *** (0.0 ± 0.0 ***)	0.3 ± 0.0 (0.1 ± 0.0)	0.2 ± 0.0 (0.0 ± 0.0)
Fruits	2.8 ± 0.5 * (0.4 ± 0.1 ^NS^)	1.0 ± 0.5 (0.3 ± 0.1)	2.0 ± 0.3 (0.3 ± 0.0)	3.2 ± 0.3 ^NS^ (0.5 ± 0.1 *)	2.8 ± 0.3 (0.7 ± 0.1)	2.9 ± 0.2 (0.6 ± 0.1)
Meats	30.0 ± 2.4 ^NS^ (4.5 ± 0.3 ***)	24.6 ± 2.5 (6.9 ± 0.6)	27.4 ± 1.7 (5.7 ± 0.3)	16.9 ± 1.6 ^NS^ (2.7 ± 0.2 ***)	20.7 ± 1.5 (5.9 ± 0.5)	18.9 ± 1.0 (4.4 ± 0.3)
Eggs	21.4 ± 1.4 ^NS^ (3.3 ± 0.2 ***)	18.1 ± 1.5 (5.7 ± 0.4)	19.8 ± 1.0 (4.4 ± 0.2)	22.9 ± 2.3 ^NS^ (3.6 ± 0.4 **)	17.2 ± 2.2 (5.0 ± 0.3)	20.5 ± 1.5 (4.3 ± 0.3)
Fish/shellfish	55.0 ± 3.2 ** (8.4 ± 0.5 **)	38.9 ± 3.3 (10.9 ± 0.7)	47.3 ± 2.3 (9.6 ± 0.4)	49.8 ± 3.6 ^NS^ (8.1 ± 0.5 ***)	46.5 ± 3.4 (11.8 ± 0.8)	46.9 ± 2.4 (10.0 ± 0.5)
Seaweed	13.3 ± 1.0 *** (2.1 ± 0.2 ^NS^)	5.7 ± 1.1 (1.7 ± 0.0)	9.66 ± 0.76 (1.9 ± 0.2)	11.2 ± 0.9 *** (1.9 ± 0.2 ^NS^)	6.1 ± 0.9 (1.5 ± 0.2)	9.0 ± 0.6 (1.7 ± 0.1)
Milk/dairy products	344.0 ± 12.8 *** (50.4 ± 1.2 ***)	132.0 ± 13.4 (24.4 ± 1.6)	242.6 ± 11.7 (38.0 ± 1.2)	314.9 ± 11.4 *** (50.0±1.2 ***)	144.5 ± 11.0 (29.7 ± 1.4)	238.7 ± 9.3 (39.5 ± 1.1)
Fat/oils	0.0 ± 0.0 *** (0.0 ± 0.0***)	0.1 ± 0.0 (0.0 ± 0.0)	0.1 ± 0.0 (0.0 ± 0.0)	0.0 ± 0.0 * (0.0 ± 0.0 ***)	0.0 ± 0.0 (0.0 ± 0.0)	0.0 ± 0.0 (0.0 ± 0.0)
Beverage	0.9 ± 0.2 ^NS^ (0.2 ± 0.0 ^NS^)	0.5 ± 0.2 (0.1 ±0.0)	0.7 ± 0.1 (0.1 ± 0.0)	0.9 ± 0.2 * (0.1 ± 0.0 ^NS^)	0.4 ± 0.2 (0.1 ± 0.0)	0.7 ± 0.1 (0.1 ± 0.0)
Seasoning/spice	16.6 ± 1.5 ^NS^ (2.5 ± 0.2 ***)	16.9 ± 1.5 (5.0 ± 0.4)	16.7 ± 1.0 (3.7 ± 0.2)	22.1 ± 1.4 * (3.7 ± 0.3 *)	17.3 ± 1.4 (4.6 ± 0.3)	19.6 ± 1.0 (4.2 ± 0.2)
Others	0.3 ± 0.1 *** (0.1 ± 0.0 ^NS^)	0.0 ± 0.1 (0.0 ± 0.0)	0.2 ± 0.1 (0.0 ± 0.0)	0.2 ± 0.1 ^NS^ (0.0 ± 0.0 ^NS^)	0.2 ± 0.1 (0.1 ± 0.0)	0.2 ± 0.0 (0.0 ± 0.0)
Source of calcium						
Plant foods	213.4 ± 6.0 ***	176.3 ± 6.3	195.7 ± 4.3	216.5 ± 6.2 ***	178.0 ± 6.0	200.3 ± 4.3
	(33.1 ± 0.9 ***)	(51.7 ± 1.4)	(42.0 ± 1.0)	(35.1 ± 0.9 ***)	(47.9 ± 1.0)	(41.7 ± 0.8)
Animal foods	451.1 ± 13.7 *** (66.9 ± 0.9 ***)	216.2. ± 14.4 (48.3 ± 1.4)	338.8 ± 12.6 (58.0 ± 1.0)	406.1 ± 11.9 *** (64.9 ± 0.9 ***)	230.8 ± 11.4 (52.1 ± 1.0)	327.1 ± 9.7 (58.2 ± 0.8)

^1^ Mean ± Standard error of mean after adjusting for age and height. ^2^ **p* < 0.05. ** *p* < 0.01. *** *p* < 0.001. NS: not significant difference (α = 0.05) ^3^ Mean ± Standard error of mean.

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
