# Peer review of "Participation in the School Milk Program Contributes to Increased Milk Consumption and Dietary Nutrient Intake by Middle School Students in South Korea"

_nutrients, 2019, doi:10.3390/nu11102386_

Round 1

Reviewer 1 Report

There are a few English grammatical errors.  The paper should be edited by a native English speaker.

Age and height should be adjusted for; things change rapidly at the age studied.   Is there a way to adjust for socioeconomic status? For example, the older and equally the taller children may eat more calories, which would explain a lot of the differences they saw in general diet.

L 78: “All subjects were informed and asked to sign the consent form for the study.”  Is the signature of a person under age 18 valid consent in Korea?  In the US we get assent from children, not consent.

L 104: “This study was conducted to compare the nutrients intake of middle school students of South Korea, by considering their participation in the school milk program. Table 1 presents the anthropometric characteristics of the study subjects.” I would omit both of the sentences.  The first was already stated and the second is effectively the title of Table 1.

Reviewer 2 Report

This is a study that evaluated the nutrient intake difference according to the participation in school milk program in South Korea. I appreciate such studies from Asian countries where dairy consumption is relatively low. However, I suggest that the overall manuscript must be improved with more interpretations, explanations and discussions.   My specific comments are below: Abstract 1) Line 18 Please change ’totally’ to ’ in total’. The same for line 74. 2) Line 21, 24 I suggest that you delete ’including calcium’ because you have not mentioned calcium in the abstract.   Introduction 3) Line 48-50 does the Guideline recommend to intake milk only for the sufficient intake of calcium? If not, please consider rewriting the sentence Line 53-55. 4) In the Introduction, I was not fully convinced about the rationale for this study: could you add some explanation about what has been done before and what should be addressed? 5) Line 65 ’Research’ is an uncountable noun. 6) I would suggest that you include ’cross-sectional’ in the objective or in the method. Also please consider to clearly state that this study is based on self-administered questionnaires, not a direct measurement in the objective or methods section.   Methods 7) I would suggest that you avoid using ’participant’ and ’non-participant’ for the description of each group because readers might be confused to see especially only the tables. 8) Line 76-78 Is there any geographical difference in these cities? Could it be possible that the two groups have some sort of influences by the geographical or social differences from where they lived? For the readers who are not familiar with South Korea, please provide more details. 9) Line 80 Which institute does ’Institutional Review Board’ mean? Please specify. 10) Line 86 It was not entirely clear what ’statistical rate’ meant. Is this the same as ’response rate’? If so, I would suggest that you change the wording. 11) Did all the students complete the record without the help of their parents? 12) I wondered if it might be better to consider the under-estimation of energy intake. Since energy intake assessment is complicated, it has been suggested to use appropriate methods when comparing to specific recommended values. For the same reason, nutrient intake might be also under-estimated. Is it reasonable to compare the raw values of the nutrient estimate based on dietary records? 13) I am not familiar with the Korean Dietary Reference Intakes, but could you add more detailed information about the comparison you conducted in this study? What kind of reference does it have? If there are multiple indices, what is the ’recommended values’ that you used? Also, I wondered how you compared the dietary intake of the students with the reference value. If you compared the mean of the intake and the reference value, was that a reasonable way to evaluate? 15) The contribution of food choices to nutrient intake was important and interesting to show. But I thought perhaps it might be helpful to show grams not percentages for this purpose.   Result 14) Overall, the tables need to be more organised. I would suggest that the footnotes 1) and 2) in each table might be better to be placed after the titles. In table 2, please consider deleting the total in each section because this is a repeating of the total number of the participants.   Discussion 14) I would like you to include more discussion about your findings. It looks to me more like a summary of previous studies. 15) Limitations of the studies and the implications for future studies would be helpful for readers.
